# Predicting Water Stress in Wild Blueberry Fields Using Airborne Visible and Near Infrared Imaging Spectroscopy

**Catherine Chan** [1,*], **Peter R. Nelson** [2], **Daniel J. Hayes** [1], **Yong-Jiang Zhang** [3] **and Bruce Hall** [4]

1   School of Forest Resources, University of Maine, Orono, ME 04469, USA; daniel.j.hayes@maine.edu
2   Schoodic Institute, 9 Atterbury Circle, PO Box 277, Winter Harbor, ME 04693, USA; pnelson@schoodicinstitute.org
3   School of Biology and Ecology, University of Maine, Orono, ME 04469, USA; yongjiang.zhang@maine.edu
4   Jasper Wyman & Son, PO Box 100, Milbridge, ME 04658, USA; bhall@wymans.com
*   Correspondence: catherine.chan@maine.edu

**Abstract:** Water management and irrigation practices are persistent challenges for many agricultural systems, exacerbated by changing seasonal and weather patterns. The wild blueberry industry is at heightened susceptibility due to its unique growing conditions and uncultivated nature. Stress detection in agricultural fields can prompt management responses to mitigate detrimental conditions, including drought and disease. We assessed airborne spectral data accompanied by ground sampled water potential over three developmental stages of wild blueberries collected throughout the 2019 summer on two adjacent fields, one irrigated and one non-irrigated. Ground sampled leaves were collected in tandem to the hyperspectral image collection with an unoccupied aerial vehicle (UAV) and then measured for leaf water potential. Using methods in machine learning and statistical analysis, we developed models to determine irrigation status and water potential. Seven models were assessed in this study, with four used to process six hyperspectral cube images for analysis. These images were classified as irrigated or non-irrigated and estimated for water potential levels, resulting in an $R^2$ of 0.62 and verified with a validation dataset. Further investigation relating imaging spectroscopy and water potential will be beneficial in understanding the dynamics between the two for future studies.

**Keywords:** hyperspectral; agriculture; vegetation indices; irrigation; machine learning; water potential; UAV; VNIR; reflectance

## 1. Introduction

The application and use of hyperspectral imaging technology to plant (stress) health and vegetation traits has made great advances in the past decade [1]. Hyperspectral imaging, or imaging spectroscopy, is a method used in detecting and classifying objects based on the light reflectance (or spectral signature) across narrowly-resolved wavelengths within the optical portion of the electromagnetic spectrum [2]. Applications range from differentiating tree species [3] to aiding agricultural quality control [4]. One of imaging spectroscopy's most valued benefits is its ability to collect information in a non-destructive manner as it does not require direct contact with the scanned object [5]. In this work, hyperspectral data analysis was used to classify irrigation status of blueberry crop areas, and determine water potential of the plant canopies.

Remote sensing has played an increasingly important role in monitoring and assessing plant health. Through its growing application (i.e., added satellite missions, sensor development, etc.) [6] and user-friendliness (i.e., programming, large data management, etc.) [7], remote sensing can provide timely and quality data over different cover types [8]. Targets of plant health include physiological responses detectable through re-

mote sensing techniques such as hyperspectral imaging [9]. These responses include nutrient declines indicating disease [10], in addition to chlorophyll content estimated through the band ratio NDVI (normalized difference vegetation index), to show sugarcane health [11]. In relation to remote sensing, imaging spectroscopy can be used as a means to estimate water potential [12].

Water management and irrigation practices are a persistent challenge for many agricultural systems in the world [13]. The effects of water stress are complex and highly influential on plant growth and productivity [14]. With many factors involved in water stress, numerous methods in quantifying water status have been developed and implemented including evapotranspiration models, soil water balance measurements, and leaf water potential [15]. Water potential is the chemical potential reduction (force that causes water movement) relative to pure water at sea level, or is a measure of the driving force of water flow [16,17] and has been shown to be an effective indicator of water stress [18,19]. Measuring water stress can further our understanding of vegetation health within natural resources, including forested lands and agricultural crops [20].

Wild blueberries (*Vaccinium angustifolium* Ait.) (also referred to as lowbush) are grown commercially in only Quebec, Atlantic Canada, and the state of Maine [21]. Lowbush blueberries require unique growing conditions including acidic and infertile soils, unsuitable for many other crop types [22]. This distinctive environment increases vulnerability due to its uncommon land makeup, further amplified by climate change factors [23,24]. In recent years, wild blueberry production has faced challenges relating to warming, drought, freezing, and pathogens [25,26]. As a result, forecasting land conditions and taking prompt mitigative action, including water management, have become increasingly needed. We analyzed the reflectance properties of blueberry crops throughout different development stages to detect patterns temporally and spatially.

Our goal was to assess the utility of a high-resolution remote sensing system to classify and predict irrigation status and water stress in blueberry crops. To address this goal, we collected hyperspectral imagery over wild blueberry fields in Downeast Maine using an unoccupied aerial vehicle (UAV), a remotely controlled aircraft also referred to as a drone [27]. From this imagery we then developed, applied, and tested machine-learning models to associate reflectance measurements to (1) a categorical response (irrigated or non-irrigated) [28] and (2) water potential as a continuous variable [29]. We planned to achieve this through:

1.  Collecting airborne data on an irrigated and non-irrigated field over three plant development stages.
2.  Acquire water potentials of canopy leaf samples in each field.
3.  Generate classified maps of irrigated vs. non-irrigated areas and estimated water potentials to determine locations of low or high plant canopy water stress.

The objectives were to configure methods and processes that will allow for more efficient detection via freely available code [30]. The process of configuration involves combining spatial measurements and ground data, in addition to approaches in UAV systems, image processing, and machine learning classifiers [31]. We sought to remotely assess irrigation status and water potentials in wild blueberry fields as a means to offer techniques in crop health monitoring.

## 2. Materials and Methods

### 2.1. Study Site

Our research was conducted on commercial blueberry fields in Deblois, Maine, owned by Jasper Wyman and Son (Wyman's), a corporation specializing in frozen fruit. Maine's coastal region has a four-season climate with an average low of −10.6 °C and high of 24.2 °C (lowest and highest month's average over 2006–2010), and monthly average precipitation low of 85.1 mm and high of 136.4 mm (lowest and highest month's average

over 1981–2010) [32]. The site includes an irrigated and a non-irrigated [33] field, approximately 23 and 16 hectares, respectively (Figure 1 displays these two fields). The irrigated field is uniformly irrigated with Nelson Full-Circle Impact sprinklers (Walla Walla, Washington, USA). Each irrigation event compensates for natural precipitation with a maximum irrigation regime of 2.5 cm of water per acre every 3 to 4 days as needed. The location is defined by Spodosols soils of the series Masardis consisting of a fine sandy loam with 0%–3% slope. These crop fields tend to contain a number of different genotypes, growing in sections within a particular field [34,35]. Due to this pattern, a number of plots were selected from the area in order to accurately assess the locations and tendencies of water potential. The irrigated and non-irrigated experimental fields are co-located adjacent to one another to reduce climate and geologic site variability and to act as a comparison between a treated and untreated site.

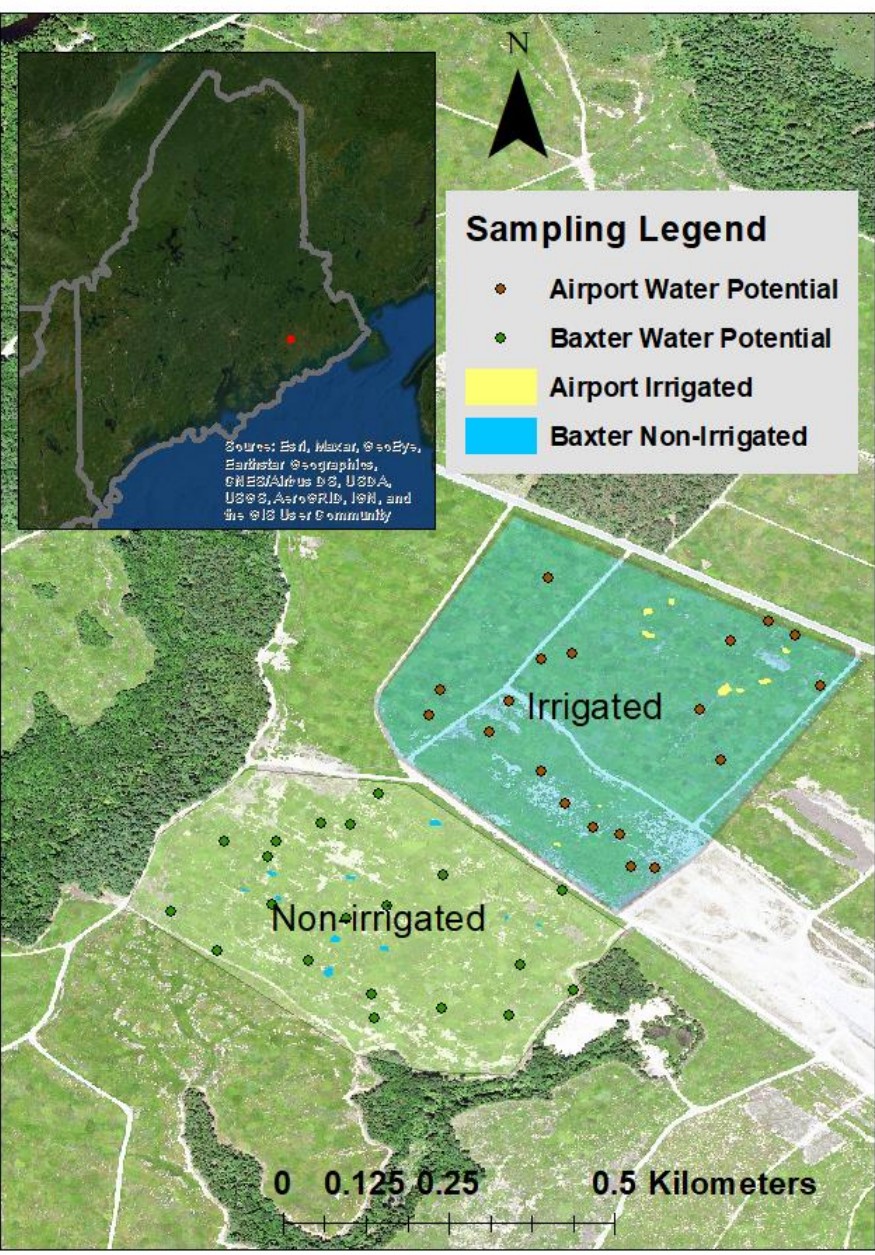

**Figure 1.** Study site of the irrigated field (upper light blue polygon) and non-irrigated field (lower light green polygon) with field sampling points and polygons.

## 2.2. Workflow Overview

The project workflow is outlined in Figure 2. Input data included the ground data (irrigated/non-irrigated classes and water potential) and the manually digitized polygons from the imagery that trained the predictor models. These points and polygons are displayed on the study site fields in Figure 1. The remainder of the nodes relate to model processing steps which are detailed further below. The models were applied to images, generating maps which were compared to validation ground measurements [36,37].

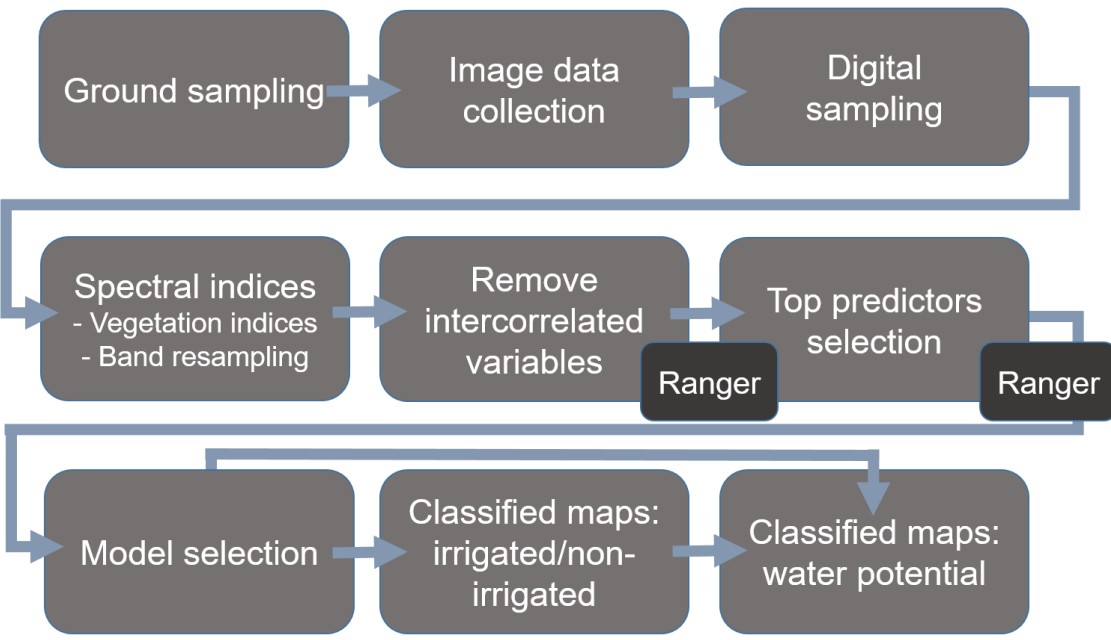

**Figure 2.** Process flow from data collection to classified and predicted images.

## 2.3. Ground Sampling

Measurements were taken over the two fields three times in the 2019 spring and summer including peak bloom (June 7), green fruit (July 3), and color break (July 25). These represent some of the development stages of wild blueberries throughout the summer growing season [38,39], where peak bloom is determined when 50% of flower buds have opened, green fruit by green color and initial fruit enlargement caused by active cell division, and color break as the period of fruits changing from green to pink to blue through cell expansion. We chose to collect data over the different stages to determine temporal variation in water conditions and spectral response over the seasonal cycle of blueberry growth [40].

We used 20 ground sample measurements on each field that were selected by a random sampling design, guided by genotypic distinction to cover spatial variations. The samples entailed a small branch of leaves attached to the stem. These were gathered as the drone captured image data. The samples were immediately stored in moist plastic Ziploc bags in coolers to minimize water loss, and measured approximately two hours later for water potentials. The water potential measurements were taken using a leaf pressure chamber (Model 1505D; PMS, Albany, OR, USA) in a climate-controlled laboratory space in Wyman's facilities.

## 2.4. Image Data Collection and Sampling

We used a Micro A-Series Sensor hyperspectral imaging spectrometer manufactured by Headwall Photonics (Bolton, MA, USA). The device is attached to a DJI Matrice 600 Pro UAV, and operates as a line-scanning (pushbroom) instrument. The sensor captures

the visible and near-infrared portion of the electromagnetic spectrum from 400 to 1000 nm and collects 324 spectral bands. The number of flight lines, heights, and speeds were determined by site area, takeoff distance from scanning area, land topography, and specific daylight conditions. Data collection with the UAV was conducted between 10:00AM and 2:00PM local time to avoid shadows in the data. Image collection was limited to these timeframes, battery capacity, and the amount of time access was allowed to the field sites.

Imagery was processed using the Headwall application Spectral View. A white tarp taken in the imagery and a spectralon panel were both considered as the white reference for image processing, however the spectralon provided more accurate spectra reflectances and greater consistency. Processing entailed transforming imagery from its raw form, to radiance, reflectance, and then orthorectification. The ground sampling distance was approximately 10 × 10 cm per pixel.

Using the collected imagery, we delineated pixels of ground sampled, or known irrigation status, blueberry leaves through the ENVI software program (version 5.5 64-bit). These delineations were extracted and used as samples, which were partitions in both the training and validation data. All downstream analyses from this point were performed in the R programming environment [30].

Training pixels in the irrigated/non-irrigated categorical model were digitized in large samples. With a binary response of irrigated and non-irrigated uninfluenced by the water potential ground reference data, we digitized large polygon areas within each field to gain a larger training size. Four samples from two images of each field were digitized for each stage.

The water potential digitizations were guided by ground reference samples. Four to eight pixels were digitized around the coordinates of where samples were collected. Pixels that were in clouded or shadowed images were discarded. Two samples were removed from the first scan date, and eight from both the second and third.

### 2.5. Model Development

A series of spectral indices were calculated from the reflectance values to use as predictors in our models [41,42]. One of these methods in deriving the variables included resampling at 5, 10, 50, and 100 nm [43,44]. These values were selected to roughly simulate wider bandpass, common in multi-spectral sensors that are commercially available. The other method of calculating vegetation indices capitalized on the 'vegindex' function from package 'hsdar' [45], which calculates almost 100 different indices depending on the bandpass of the data. The resampling and vegetation indices totaled 260 variables as predictors.

A random forest model (classifier which uses decision trees) was trained to classify images [46] with the calibration data (either water potential or irrigated/non-irrigated) using the 'ranger' package to determine the relationship between the water potentials of ground measurements and the spectral index predictors. Ranger was selected rather than the 'Random Forest' package due to its speed and improvements in variable importance measures and bias [47].

The process of developing the models (to classify and predict images) first entailed removing intercorrelated variables among all predictors using the 'caret' package in R through Pearson correlation coefficient [48]. After removing these variables, we applied ranger to determine which predictors are most important. Then, we again used ranger on the top 50 important variables. A list of models was produced, each utilizing a different number of the most important variables with accuracy rates. We selected the model that utilized the fewest number of variables, but produced a low error rate.

Prior to final model selection and image classification, we performed a validation test on the water potential models. We redeveloped the models with a split of 70% calibration and 30% validation from the total sample set, randomly chosen from each field. The split was chosen due to the small sample sizes with peak bloom having 139 pixel samples, green fruit 115, and color break 99. The global model prediction, however, used an 80% and 20% split of the total sample data due to the combined sample sizes.

## 3. Results

We developed models that used the extracted pixel samples and water potential measurements as inputs to predict characteristics of unknown areas of imagery [49]. Using functionality developed by related projects, the products of the models were generated maps of irrigated or non-irrigated areas, and water potential classifications [30].

The imagery dataset of irrigated and non-irrigated fields consists of 48 images collected in the three phenological stages, with a spatial resolution of approximately 10 cm. Ground referenced water potentials were measured as described above. One section on the irrigated field experienced winter damage, leading to a duplicated sample to detect potential differences.

### 3.1. Model

Different models were developed from the digitized samples of known conditions [50]. Two models were created for each field stage, with one as a binary classification (irrigated or non-irrigated), and the other a continuous estimation (water potential). Additionally, a global model for water potential was trained with combined samples from all three stages. Table 1 outlines the model information and error rates.

Model fit improved when highly intercorrelated predictors were removed. Errors in removing intercorrelated predictor variables were calculated at 0.9, 0.93, 0.96, and 0.99 pair-wise absolute correlation cutoff levels. Removing these at a 0.99 cutoff produced the highest accuracy rates.

Each model's sample size was the number of digitized pixels. The categorical models had out-of-bag (OOB) errors from 0.003% to 1.402%. The stage-specific (or local) models for water potential had an $R^2$ ranging from 0.437 to 0.487. The global model utilized all pixel samples as training data and resulted in an $R^2$ of 0.554. For further analysis, predictions were also conducted for all four of these water potential models.

**Table 1.** Model information used in image classification with each collection stage having a separate model, and combined stages for global model regression. The table also includes out-of-bag prediction errors as percentages for classification, and mean squared error and $R^2$ values for regression.

| | Irrigated/Non-Irrigated Local Classification | | | Water Potential Local Regression | | | Water Potential Global Regression |
|---|---|---|---|---|---|---|---|
| | Peak Bloom | Green Fruit | Color Break | Peak Bloom | Green Fruit | Color Break | |
| Sample Size (pixels) | 47,758 | 32,018 | 103,135 | 139 | 115 | 99 | 353 |
| Independent variables | 25 | 25 | 25 | 15 | 25 | 25 | 20 |
| Out-of-bag (OOB) prediction error/MSE | 1.346% | 0.003% | 1.402% | 320 | 679 | 754 | 709 |
| $R^2$ (OOB) | NA | | | 0.487 | 0.458 | 0.437 | 0.554 |

### 3.2. Validation

To further analyze the performance of the continuous models, predictions using calibration and validation samples were conducted. The increased split ratio for calibration data in the global water potential model was increased because it improved the model's

R² while still maintaining a validation size of about 70. Table 2 outlines each model's OOB error, R², and calculated RMSE.

**Table 2.** Outline of global and local regression model statistics for predicting water potential.

| Validation Metric | Local Regression Prediction | | | Global Regression Prediction |
|---|---|---|---|---|
| | **Peak Bloom** | **Green Fruit** | **Color Break** | |
| OOB prediction error (MSE) | 237 | 533 | 556 | 616 |
| R² (OOB) | 0.554 | 0.563 | 0.564 | 0.617 |
| Calculated RSME | 29.8 | 36.9 | 45.2 | 46.4 |

The table shows that all three local models had very similar R² values although calculated RMSEs were dissimilar and increased with each model. The global model had a comparable RMSE to the third local model but also had the highest R². As a result of the prediction, we decided to use only the global water potential model for our image classifications.

### 3.3. Variable Importance

Each model was developed through particular predictor variables and a certain number of the top predictors [51]. These predictor variables, or spectral derivatives, consist of vegetation indices and resampled bands. Models of differing numbers of top predictors (in multiples of five) were generated in the process along with accuracy rates, however the one with a lower number of variables with a comparable lower error rate was selected to improve efficiency but maintain efficacy. The plots of the top predictors show the diminishing importance of each variable and how, despite there being a level of relevancy, inclusion of more variables does not improve model accuracy.

Figure 3 is a plot of the 35 most important predictor variables in the global model with the importance levels, and Table 3 lists the top 20 predictors that were used in the model. Bandpasses are expressed with an 'X' preceding the bandpass wavelength, followed by the resampling range. All of the top five predictors involve bands in the 700 to 800 nm region, a range associated with chlorophyll measurements. The top predictor was the Datt index, which was developed to estimate chlorophyll content, particularly in higher plants or tree canopies [52]. The Giltelson index is a measure of chlorophyll fluorescence, which is proportional to actual chlorophyll content [53]. The third most important variable was TOCARI2OSAVI2, which incorporates soil adjustment in chlorophyll measurements and has been recommended for agricultural applications [54].

In the supplementary, Figures S1 and S2 show the top predictor variables for each model of local classification and regression. The top predictors of the three categorical models showed very distinct variable selection results with no model containing a shared top five variable. The first shared predictor was CRI2 as the fifth for color break and the eighth of green fruit. This predictor was the 24th for peak bloom. Green fruit and color break had predictors focused on chlorophyll whereas the top predictor of peak bloom was bandpass X897.593 resampled at 100 nm. In all three individual models for water potential, MTCI (MERIS Terrestrial Chlorophyll Index) was a high predictor. Medium Resolution Imaging Spectrometer (MERIS) terrestrial chlorophyll index measures chlorophyll in the red edge region distinguishing it from other red edge position indices by its sensitivity to higher levels of chlorophyll [55]. The local regression plots are included in the supplementary materials to show the similarities and differences among local predictors, and in comparison to the global model.

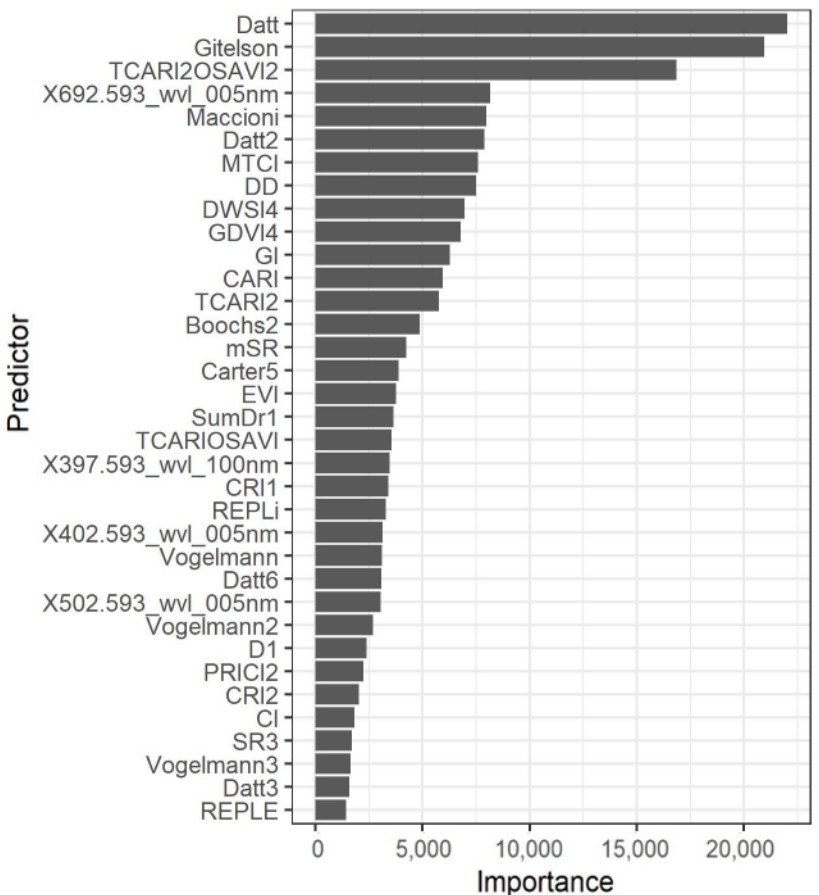

**Figure 3.** Chart of the 35 most important predictor variables in the global water potential model, the first 20 of which were included in the selected classifier, and corresponding importance levels.

**Table 3.** Top 20 predictor variables used in the global water potential model with description and band formula.

| | Abbreviation | Name | Formula |
|---|---|---|---|
| 1 | Datt | 'Chlorophyll & height' | $(R_{749} - R_{720}) - (R_{701} - R_{672})$ |
| 2 | Gitelson | 'Chlorophyll' | $1/R_{700}$ |
| 3 | TCARI2OSAVI2 | Transformed Chlorophyll Absorption Ratio 2/Optimized Soil Adjusted Vegetation Index 2 | $(3 * ((R_{750} - R_{705}) - 0.2 * (R_{750} -R_{550}) * (R_{750}/R_{705})))/ ((1 + 0.16) * (R_{750}-R_{705})/(R_{750} + R_{705} + 0.16))$ |
| 4 | X692.593_wvl_005 nm | | 'Bandpass 692.593 resampled at 5 nm' |
| 5 | Maccioni | 'Chlorophyll' | $(R_{780} - R_{710})/(R_{780} - R_{680})$ |
| 6 | Datt2 | 'Chlorophyll & height' | $R_{850}/R_{710}$ |
| 7 | MTCI | MERIS Terrestrial Chlorophyll Index | $(R_{754} - R_{709})/(R_{709} - R_{681})$ |
| 8 | DD | Double Difference Index | $(R_{749} - R_{720}) - (R_{701} - R_{672})$ |
| 9 | DWSI4 | Disease Water Stress Index 4 | $R_{550}/R_{680}$ |
| 10 | GDVI4 | Green Difference Vegetation Index 4 | $(R^4_{800} - R^4_{680})/(R^4_{800} + R^4_{680})$ |
| 11 | GI | Greenness Index | $R_{554}/R_{677}$ |

| 12 | CARI | Chlorophyll Absorption Ration Index | $R_{700} * abs(a * _{670} + R_{670} + b)/R_{670} * (\alpha^2 + 1)^{0.5}$ $\alpha = (R_{700} - R_{550})/150$ $b = R_{550} - (550 * \alpha)$ |
|----|------|------|------|
| 13 | TCARI2 | Transformed Chlorophyll Absorption Ratio 2 | $(3 * ((R750 - R705) - 0.2 * (R750 - R550) * (R750/R705)))$ |
| 14 | Boochs2 | Single Band 703 Boochs | $D_{703}$ |
| 15 | mSR | modified Simple Ratio | $(R_{800} - R_{445})/(R_{680} - R_{445})$ |
| 16 | Ctr5 | Carter 5 | $R_{695}/R_{670}$ |
| 17 | EVI | Enhanced Vegetation Index | $2.5 * ((R_{800} - R_{670})/(R_{800} - (6 * R_{670}) - (7.5 * R_{475}) + 1))$ |
| 18 | SumDr1 | 'LAI & % green cover' | $\sum_{i=626}^{795} D1i$ |
| 19 | TCARIOSAVI | Transformed Chlorophyll Absorption Ratio/Optimized Soil Adjusted Vegetation Index | $(3 * ((R_{700} - R_{670}) - 0.2 * (R_{700} - R_{550}) * (R_{700}/R_{670})))/ ((1 + 0.16) * (R_{800} - R_{670})/(R_{800} + R_{670} + 0.16))$ |
| 20 | X397.593_wvl_100 _nm | | 'Bandpass 397.593 resampled at 100 nm' |

A component of this study assessed variable importance for each model to understand how our imaging spectroscopy methods provide information. Most top predictor variables in the models focus on assessing chlorophyll levels. Some, however, include elements of soil adjustment such as TCARI2OSAVI2 and REPLi. Most of the vegetation indices and relevant bands range between 650 to 800 nm on the electromagnetic spectrum, or in the visible to near infrared region, which is displayed in the reflectance spectra (Figure 4).

### 3.4. Spectral Signatures

Various spectra of blueberry samples and pixels were observed to assess radiometric calibration and provide visual representation. The critical analysis, however, was using these spectra in development of the spectral variables, being the calculations of the vegetation indices and band resampling values. Nonetheless, the signatures can showcase information of interest. Figure 4 plots the median value signatures of combined samples, one from each field over each development stage. As can be seen in the figure, the spectra of peak bloom in the shades of pink closely align in the 500 to 700 nm range, but diverge after where the reflectance of the non-irrigated field shows higher than the irrigated. The reflectance of green fruit (shades of green) and color break (shades of blue) differ from the first stage where the irrigated signatures have higher reflectance rates than the non-irrigated, although the spectra of color break have similar values in the higher section of the spectrum. The difference of the irrigated and non-irrigated spectra of green fruit are also much larger than that of color break. These median reflectance spectra display the collected data in a raw form, but the transformations of these is what the model is developed from, and used to classify imagery.

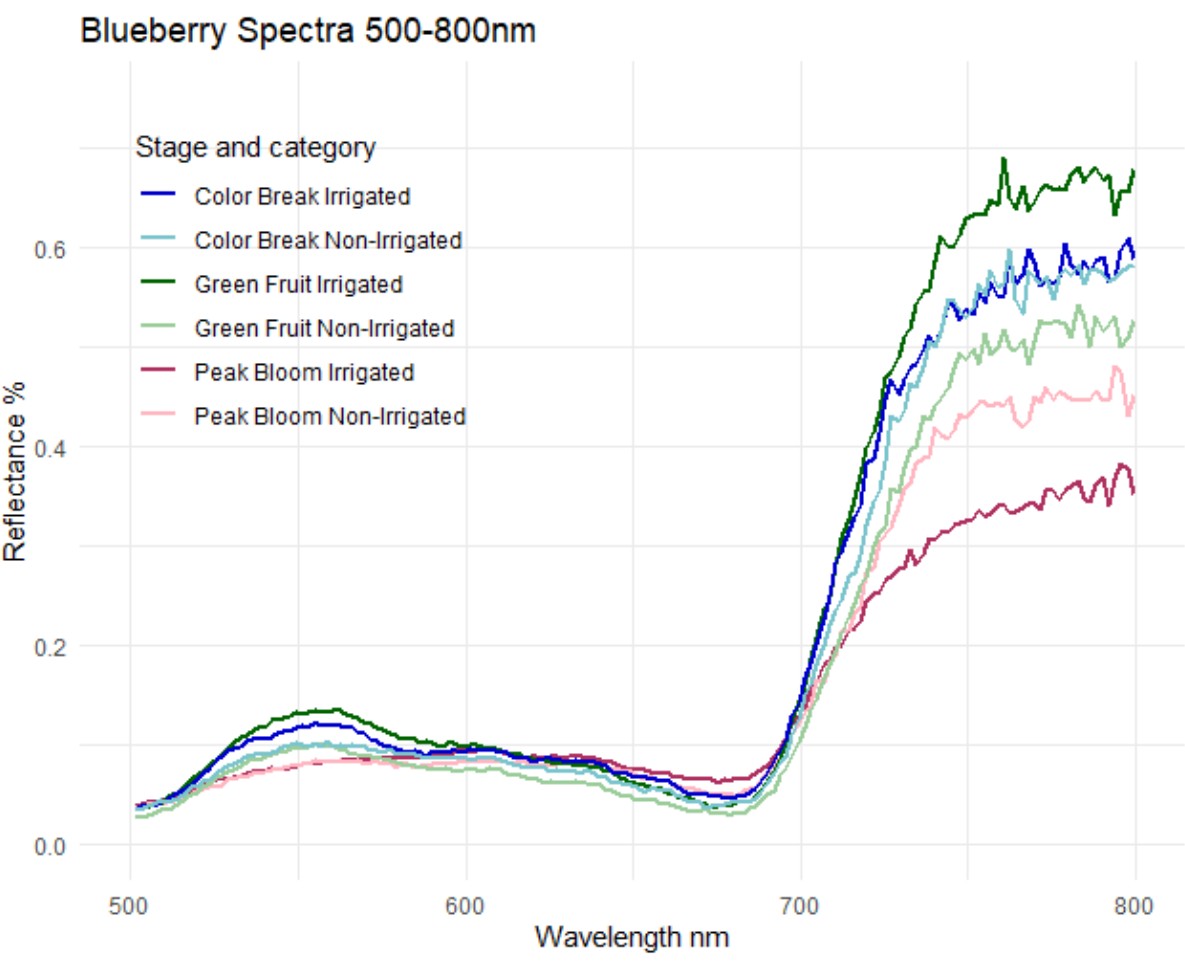

**Figure 4.** Spectral plots of median reflectance for all sampled blueberry pixels from each field and stage.

### 3.5. Classification

Cube images of the same area that were well-illuminated for all three stages were classified for analysis which amounted to a total of 12. Figures 5–7 each represent a separate stage, or date the image was taken with an irrigated image on top, and non-irrigated image on bottom. They first display NDVI images of the two sections, followed by a classified image as irrigated or non-irrigated, and then a predicted image of water potentials. The cube images from the different stages are of the same area and extent on the irrigated and non-irrigated fields.

All categorical images were classified accurately, meaning a majority of each image was classified as it actually was. All irrigated images were classified correctly, and the same result occurred with the non-irrigated. The global water potential model estimated large portions of the irrigated fields to have higher values than the non-irrigated fields of the same stage. This was especially evident in the peak bloom stage, which is consistent with the conditions of that year as precipitation was high that spring. The collection date also took place later in the season than scheduled due to unanticipated rain dates, resulting in the collected data being more representative of the transition stage between peak bloom and green fruit. The images are also shown to decrease in values throughout the season, or become more stressed throughout the summer.

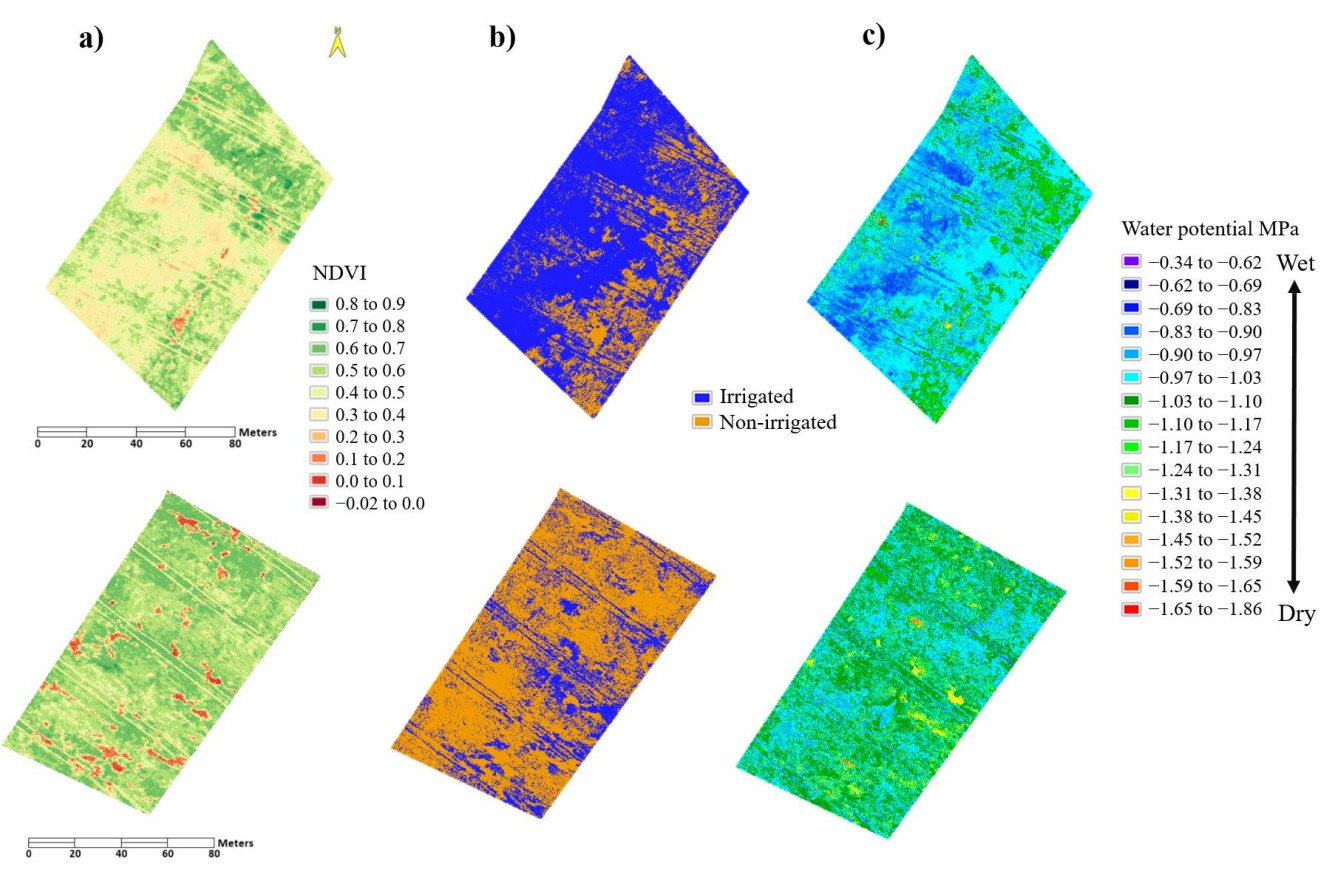

**Figure 5.** Processed scene images of irrigated (top images) and non-irrigated (bottom images) field sections over peak bloom. (**a**) (normalized difference vegetation index (NDVI) images of scenes. (**b**) Classified images of irrigated and non-irrigated scenes. (**c**) Predicted images of water potential.

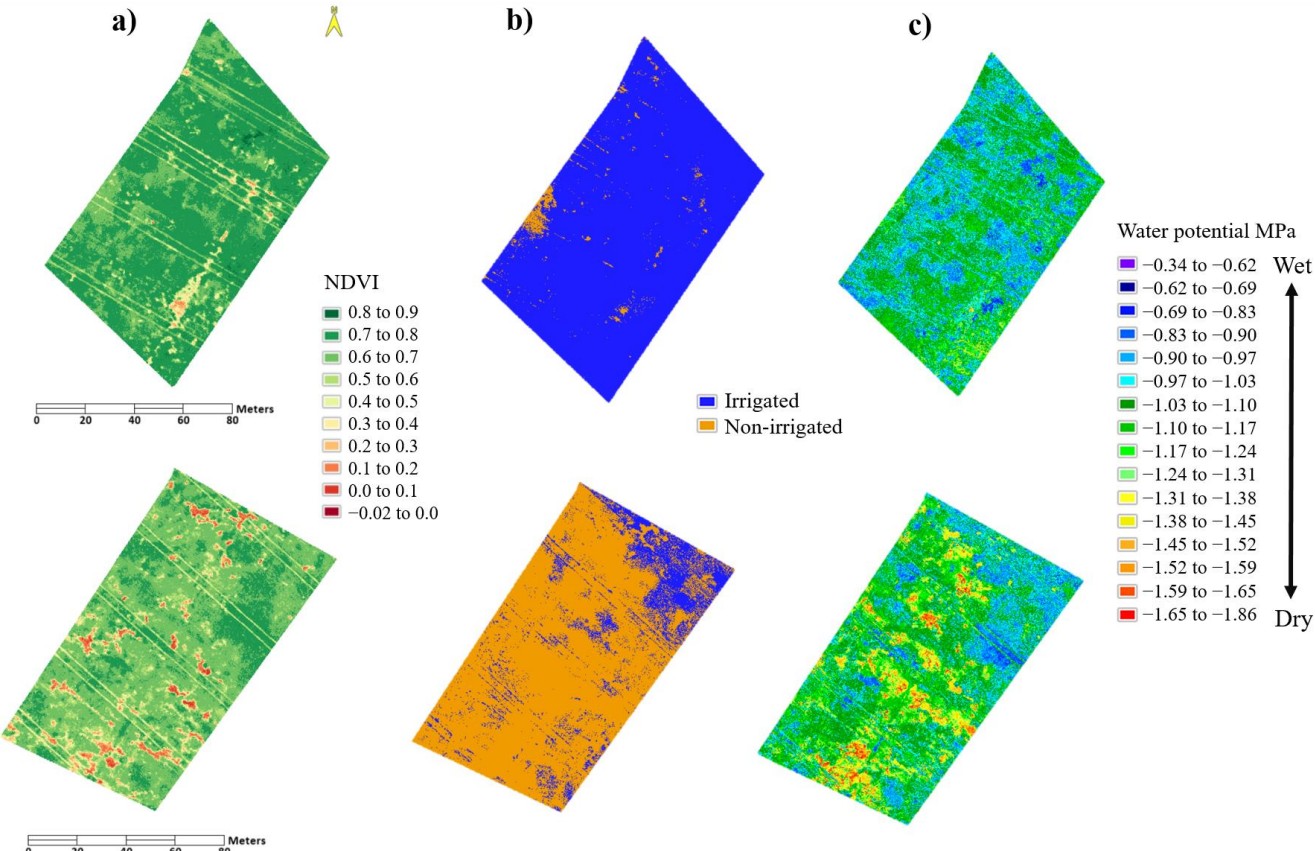

**Figure 6.** Processed scene images of irrigated (top images) and non-irrigated (bottom images) field sections over green fruit. (**a**) NDVI images of scenes. (**b**) Classified images of irrigated and non-irrigated scenes. (**c**) Predicted images of water potential.

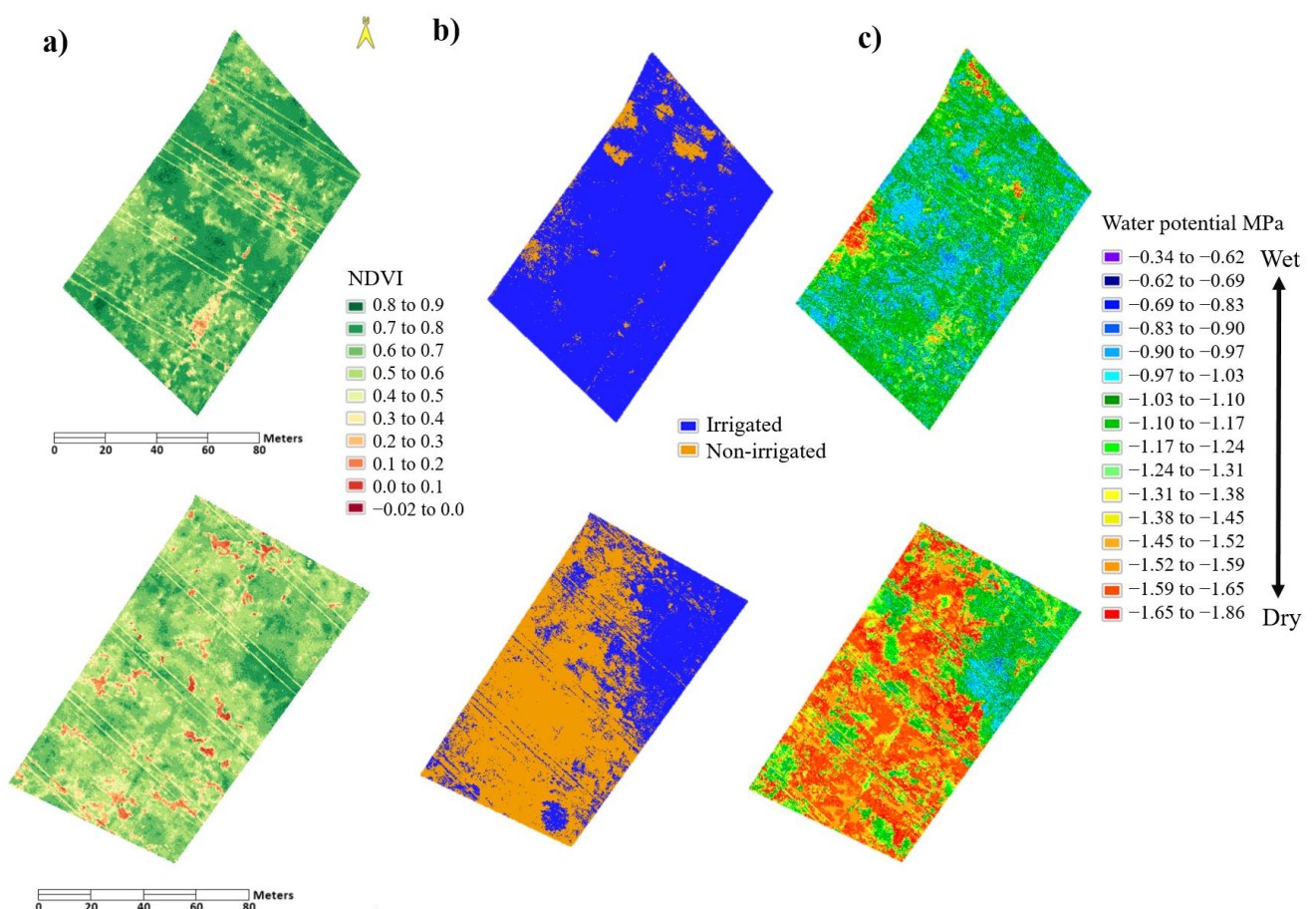

**Figure 7.** Processed scene images of irrigated (top images) and non-irrigated (bottom images) field sections over color break. (**a**) NDVI images of scenes. (**b**) Classified images of irrigated and non-irrigated scenes. (**c**) Predicted images of water potential.

## 4. Discussion

This study utilized machine learning and remote imaging spectroscopy to predict water irrigation status and water potentials in wild blueberry fields. The spectral indices produced using the ranger package were found to be effective as predictor variables in classifying hyperspectral cube images, where the indices used had a logical premise and displayed spectral sensitivity through band transformations. We analyzed our developed models to understand how the results were predicted.

The predictors in the local categorical models showed more variation among the three than those predicting local water potentials. This result is likely due to the prediction method in which the binary model target was more generalized than the continuous. Assigning a broad binary response to the large input sample data may have been too encompassing for a diverse sample, leading the model to use varying predictor variables [56]. Conversely, a specific water potential value assigned to fewer input pixels could have resulted in more refined and distinct predictor variables. Predictors in the local water potential models were found to be less variable than those of the categorical models, however both of these sets also showed differences as a result of the development stages.

The models in each set of local predictors exhibited variation, potentially a result of developmental growing patterns and a reflection of the distinct characteristics of each stage used for estimation. Our observations provide a framework in which each date represented a stage, where peak bloom is characterized as healthy and lush vegetation but lower leaf area, green fruit as higher vegetation area with additional purples, pinks, and whites from flowerings, and color break expressing darker shades from fruit maturity and

the start of senescing leaves. The predicted images of water potential display temporal variance between the blueberry development stages, and both categorical and continuous models showed distinction in statistics and outputs. This was shown particularly between the first stage and the following two of the categorical models. These observations are supported through peak bloom having more distinguishable phenology from the other two, where leaf area was much smaller and ground soils were relatively more exposed [57]. Capturing the bare ground may have been a factor in the soil adjusted indices that were relevant in the first blueberry scanning [58]. In contrast, green fruit and color break had more vegetation coverage which can be observed in the NDVI images of Figures 5–7 (with the exception of the non-irrigated image of the color break stage, possibly a result of drought stress or other factors). Due to the respective training inputs, differences between local models are to be expected [59], however the differences between local models and one that is global can be analyzed to understand how targets are estimated.

The global model used input data from all three development stages as training data, differing from the local models which used only those data from its respective date. Its estimation method was the result of a consolidation of characteristics from peak bloom, green fruit, and color break, resulting in important indices that will account for the variability [59]. It can be observed that the top three predictors of the global model had a significant level of importance over the following ones, or in other words, the importance level of the fourth predictor is much lower than the third. This differs from the local models where the levels had a more gradual decline in importance. The diverse training samples may have resulted in generalized predictors that emphasize traits related to chlorophyll [60,61], rather than a specific index, such as the first predictor of the local water potential model of color break, which was plant senescence reflectance index (PSRI) [62]. Understanding the manner in which the global water potential model makes estimates provides guidance in interpreting the predicted images.

The predicted water potential images for each field in each development stage show variations that correspond to conditions of that field or season. As the season progressed, estimated images exhibit increased water stress in all scenes. The irrigated fields show to be less stressed than the non-irrigated, supporting the irrigation management practices. This does however raise the question of what conditions are most conducive to blueberry yield, and what stresses might be permissible [63]. Understanding the baseline levels of water potential for each stage is critical in determining favorable or damaging conditions [64]. Typically, we have observed that the irrigated field generates a much higher yield than the non-irrigated, however in 2019, the irrigated field only produced a small amount more. The cause of this decreased yield however, was determined to be winter damage from a consistent freeze in the irrigated field [65]. Overall, the water potential model was found to estimate well across the development stages. The categorical models of predicting irrigated and non-irrigated also classified accurately, however these results raise the question of how useful the information actually is.

An objective of this study was to associate reflectance measurements with a categorical treatment (irrigated or non-irrigated) to make predictions. This was achieved, however there were areas that were misclassified. These misclassifications may show the regions of the field that are particularly water stressed in the irrigated field, or water sufficient in the non-irrigated field. Regardless, the method of classifying areas as irrigated or non-irrigated prompts questioning about how constructive the answer actually is. The local models were separated using only data taken from the respective date. This was due to the relativity in deeming a field irrigated or not irrigated throughout a changing season [66]. An area classified as irrigated at the beginning of the season may not be classified the same towards the end. The irrigated/non-irrigated distinction is a convenient method in classification and may be a useful precursor to mapping water potential, but finding utility in it alone may be a challenge.

A more sensitive approach is to predict and calculate water potential which provides relevant values [67]. Measuring water potential and how it would impact irrigation practices is a more appropriate pursuit in providing farmers and landowners greater use. Although the global water potential model predicted well against the validation dataset ($R^2$ of 0.62), adding more samples in water potential and from more than a few select dates would assist in more applicable predictions. Another addition in strengthening the model and prediction process would be to include other types of variables from other functions or calculations. Spectral band derivatives and extensions of these, another common method in determining characteristics of vegetation [68], were not heavily incorporated in this process, with the exception of a few vegetation indices that include derivatives in the index formula. It may also be helpful in reducing the resampling predictors as it may not have had a significant effect on the predictors, especially as the number of band resampling increased.

In comparison to other studies that have used hyperspectral data and vegetation indices to predict crop water potential, this study shows similar promise in predictive ability [69–71]. In Zhang et al. (2019), it was found that select vegetation indices were sensitive to water status variables including canopy water content, but would benefit with additional data on biomass or vegetation structure [72]. With the more rudimentary methods in our statistical analysis and classifier model (as opposed to Pôças, 2017) [73], our study further supports spectral vegetation indices through utilizing a high number of predictors. It also contributes to predictive efforts in producing maps which show spatial variation on smaller scales. Given the finer resolution, this study's application is mainly practical for limited areas, such as agricultural fields. Our methods however, were developed for efficient and applicable use.

## 5. Conclusions

The goal of this project was to use hyperspectral imaging processes in detecting water stress level practically, for the benefit of the agricultural industry. This was mainly achieved through the development of our models, although there are a number of actions that could strengthen the outcomes and better show the relationship between imaging spectroscopy and leaf water potential measurements. Models will be improved if larger training datasets are used. This is particularly relevant if other blueberry fields or varying crops are to be measured, such as fields in other areas. The process of streamlining by means of data collection, processing, and efficient programming, is also critical to provide timely and accurate data that can be efficiently used by the industry, which would require greater computing capacity and data management. With continued input additions and modifications, our methods can assist in improved agricultural practices.

Many societal structures are heavily reliant on natural and agricultural resources [74]. With the limited commercial production of wild blueberries, efficient practices and maintenance in the face of climate change are critical [24,26,75]. New technologies such as those pertaining to precision agriculture are becoming more widely used [76], necessitating the adoption of new methods to maintain techniques in a changing environment. With greater sampling and in-depth studies, hyperspectral imaging methods could have many suitable applications.

**Supplementary Materials:** The following are available online at www.mdpi.com/2072-4292/13/8/1425/s1, Figure S1. Plot of the 35 most important predictor variables in the local models and corresponding importance levels, Figure S2. Plot of the 35 most important predictor variables in the local water potential models and corresponding importance levels.

**Author Contributions:** Conceptualization, C.C. and P.R.N.; methodology, C.C., P.R.N. and D.J.H.; software, P.R.N.; validation, C.C., P.R.N. and D.J.H.; formal analysis, C.C., P.R.N.; investigation, C.C., P.R.N.; resources, P.R.N., Y.-J.Z. and B.H.; data curation, C.C., Y.-J.Z.; writing—original draft preparation, C.C.; writing—review and editing, C.C., P.R.N., D.J.H., Y.-J.Z., B.H.; visualization, C.C.,

P.R.N.; supervision, P.R.N. and D.J.H.; project administration, P.R.N.; funding acquisition, P.R.N. All authors have read and agreed to the published version of the manuscript.

**Funding:** This research was funded by the Maine Economic Improvement Fund (MEIF) funded by the state of Maine as well as the Research Reinvestment Fund (RRF) through the University of Maine. This project was supported by the USDA National Institute of Food and Agriculture, Hatch (or McIntire-Stennis, Animal Health, etc.) Project number ME0-41907 through the Maine Agricultural & Forest Experiment Station. Maine Agricultural and Forest Experiment Publication Number 3806.

**Institutional Review Board Statement:** Not applicable.

**Informed Consent Statement:** Not applicable.

**Data Availability Statement:** All code is available at the following two Github pages: https://github.com/nelsopet/lecospec (accessed on 28 February 2021) and https://github.com/catherinechan70/Blueberries (accessed on 28 February 2021), please use correspondence for any additional inquiries.

**Acknowledgments:** We acknowledge the land used and studied in this project is occupied territory of the native Penobscot and Passamaquoddy peoples. Additionally, we would like to thank field crew members Pratima Pahadi, Arin Chen, Aldous Hofmann, Sarah Marcotte, and Valeria Briones for help with field sample collection.

**Conflicts of Interest:** The authors declare no conflict of interest. The funders had no role in the design of the study; in the collection, analyses, or interpretation of data; in the writing of the manuscript, or in the decision to publish the results.

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
