# Peer review of "Predicting Water Stress in Wild Blueberry Fields Using Airborne Visible and Near Infrared Imaging Spectroscopy"

_remotesensing, doi:10.3390/rs13081425_

Round 1
Reviewer 1 Report
This paper is well-written and structured. Study results may add to the existing knowledge. However, the following comments may enhance the readability of this paper:
- All acronym names such as UAV, RF, The three should be defined when first appear.
- The last sentence (Lines 21-22) in the abstract has not been discussed in the main text.
- The objectives and goals of this study may be described in a more concise manner.
- Some details on the basic conditions (soil, topography, geology, and climate) of the irrigated and non-irrigated field may be helpful.
- The three developmental stages of blueberry should be defined in more detail.
- In Line 177, the irrigated field area is 23 ha. However, in Line 93, the field area has been mentioned as 16 ha. Why?
- Values for the last row of Table 1 are missing?
- How to estimate “importance levels” (in Figures 4-6) in order to determine variable importance?
- There are altogether 35 predictors. What are they? How to come with all these predictors?
- Please explain the statement in Lines 377-378.
Reviewer 2 Report
The work aims to assess the utility of a high-resolution remote sensing system to monitor irrigation and water stress in blueberry crops. The authors develop some models based on machine learning to predict the water potential from some vegetation indices. The paper is of great value and contributes significantly to the research within remote sensing field. However, it is not clear which type of models have been developed? I guess that the models are based on the formulas of table 3, but the equations between different vegetation indices and water potential are missing.
Some other comments below:
- Add “machine learning” in the title
- Line 34-35. Replace “in this project” by “in this work”
- I propose for the authors to move figure 4 and table 3 as supplement materials
- I think “Section 3: Results” is missing after line 172
- Figure 8,9,10 (a). The color bar (legend) for NDVI images is missing.
Reviewer 3 Report
The subject of this paper is interesting because shows a new methodology to characterize water stress in a crop that is important for the North American market.
However, I had some difficulties reading the paper and understanding how the model implemented works. The explanation of the processing chain is incomplete; the description of the single modules of the “ranger model” and how they are implemented and validated are the core of this paper, but they are not clearly and sequentially described (starting from the flow chart and, consequently, to the sub-chapters organization), thus making difficult the replicability of the experiment with other crop types or over different climatic areas. The structure of chapters and sub-chapters is not well defined: there is no separation among the methodology and results; some sub-chapters could be unified, others should be reorganized and their role in the model should be better explained; redundancies could be avoided in order to ensure an easier reading.
The irrigated/non-irrigated classification used to define the output of one of the models is misleading, because this terminology is referred to the two selected Airport-irrigated and Baxter non irrigated study areas (see comment n.18).
In the introduction, updated references should be included.
Other comments related to the paper are:
- Line 16: Please, specify "leaves water potential"
- Lines 92-93: What irrigation system was used in this blueberry field? It should be important to know it in order to have an idea of the water distribution and its efficiency. Please move in this section information concerning the extension of the two fields considered for the study (see lines 177-178: "The imagery dataset used in this study includes 23 hectares over the irrigated field 177 and 16 over the non-irrigated field...")
- Sub-chapter 2.2 – "Workflow overview": Figure 2 is not a process flow, but a simple description of the main steps done. It could be better to include some of these steps (remote sensing and ground data collection) in Figure 3, in order to have only one comprehensive and detailed workflow of the whole system. Therefore, starting from the complete workflow, you could describe in detail the sections of Digitization, Training, Validation, and Final application to the whole fields. Figure 3 can be moved in this sub-chapter but must be modified. The workflow, in fact, is incomplete and confusing and it should be useful to use basic flowchart symbols. Please, add the data collection (see the comment of Figure 2) specifying in which step remote sensing and ground data are used; detail the training and validation steps; better collocate the resampling and the vegetation indices calculation; reorder the light-green boxes into the workflow between the random forest and the classified output boxes. Through this flow chart readers should easily understand the general approach and the whole system and starting from the flow chart, you can better organize the following sub-chapters.
- Sections 2.3 and 2.5: these two sections could be merged into "Image data collection and sampling"
- Line108: Please specify that it is a hyperspectral camera.
- Lines 124-125: Please, could you include figures of the location of polygons used as training areas? What do you mean by "phenological distinction"? Is it the fraction of vegetation cover or what else? Or did you refer to different genotypes mentioned at line 90? Genotype and phenology are two different concepts, so what is the principle chosen for selecting sample areas?
- Reference 44: please, check and update the reference: Lehnert, L. W., Meyer, H., Obermeier, W. A., Silva, B., Regeling, B., & Bendix, J. (2019). Hyperspectral data analysis in R: the hsdar package. J. of Statistical Sotware. 89(12). doi: 10.18637/jss.v089.i12
- Line 166: "...between the water potentials of ground measurements and the spectral index predictors."
- Reference 46: please, check and update the reference: Wright, M. N., & Ziegler, A. (2017). ranger: A fast implementation of random forests for high dimensional data in C++ and R. J. of Statistical Software. 77(1). doi: 10.18637/jss.v077.i01
- Lines 168-176: You use the general terms "models", "model" and again "models". Are "models" mentioned in lines 168-170 the same as lines 173-176? Please, specify "Ranger model" each time you refer to it instead of a general "model" to avoid confusion with the other "models".
- Lines 174-175: check the English
- Lines 177-179: Some information is redundant. The dimensions of the fields are defined in the “study site” sub-chapter. Simplify the sentence: "The imagery dataset of irrigated/non-irrigated fields consist of 48 images collected in the three phenological stages, with a spatial..."
- Lines 179-180: This sentence is redundant (see lines 124-125).
- Line 186: Check the English
- Table 1: please include all the missed measure units in the table and the caption.
- Lines 287-292: You are mentioning here two areas with different names and water management (Airport/Irrigated and Baxter/non-irrigated), but in the “Study area” section areas are differently described. Are they the same? if yes, homogenize the names. Are these Images a portion of the fields highlighted in Figure 1? Specify it in the captions.
- Figure 8-9-10: Please, include the NDVI legend.
- “Discussion” chapter: The categorical definitions of irrigated/non-irrigated classes are misleading. Based on your field’s description, you did the study over two fields with different water management (i.e. Airport is irrigated and Baxter is not irrigated). So, when your model identifies irrigation pixels in a non-irrigated field and vice versa (Figg. 8-9-10), what they really mean? In the case of the Baxter non-irrigated field, what is the mining of the "irrigated" class? And what the meaning of the "non-irrigated" class in the Airport irrigated field?
Round 2
Reviewer 2 Report
The paper has been celearly improved and all my comments have been adressed. Then I recommend publication in actual form.
Reviewer 3 Report
The authors took my comments into account and improved the work in line with my recommendations. I recommend the paper for publication.